# Psychosocial Interventions for Attention Deficit/Hyperactivity Disorder: A Systematic Review and Meta-Analysis by the CADDRA Guidelines Work GROUP

**DOI:** 10.3390/brainsci12081023

**Published:** 2022-08-01

**Authors:** Valerie Tourjman, Gill Louis-Nascan, Ghalib Ahmed, Anaïs DuBow, Hubert Côté, Nadia Daly, George Daoud, Stacey Espinet, Joan Flood, Emilie Gagnier-Marandola, Martin Gignac, Gemma Graziosi, Zeeshan Mansuri, Joseph Sadek

**Affiliations:** 1Department of Psychiatry, Université de Montréal, Montreal, QC H3T 1J4, Canada; valtour@outlook.com (V.T.); gill.louis-nascan@courrier.uqam.ca (G.L.-N.); anaisd618@hotmail.com (A.D.); hubert.cote.3@gmail.com (H.C.); 2Department of Psychology, Université du Québec à Montréal, Montreal, QC H2X 3P2, Canada; 3Departments of Family Medicine and Psychiatry, University of Alberta, Edmonton, AB T6G 2R3, Canada; ghalib@shaw.ca; 4Department of Psychiatry, Harvard University, Boston, MA 02115, USA; nadia.daly@mail.mcgill.ca; 5Boston Children’s Hospital, Harvard University, Boston, MA 02115, USA; zeeshanmansuri@gmail.com; 6Department of Psychiatry, Université de Sherbrooke, Sherbrooke, QC J1K 2R1, Canada; george.daoud@usherbrooke.ca; 7CADDRA—Canadian ADHD Resource Alliance, Toronto, ON M5A 3X9, Canada; stacey.espinet@caddra.ca; 8Department of Psychiatry, The Shoniker Clinic, Scarborough, ON M1E 4B9, Canada; joanmflood@gmail.com; 9Department of Family Medicine, McGill University, Montreal, QC H3A 0G4, Canada; emilie.gagnier-marandola@mail.mcgill.ca; 10Child and Adolescent Psychiatry Division, McGill University, Montreal, QC H3A 0G4, Canada; martin.gignac.med@ssss.gouv.qc.ca; 11Department of Psychology, York University, Toronto, ON M3J 1P3, Canada; gemmagraziosi@gmail.com; 12Department of Psychiatry, Dalhousie University, Halifax, NS B3H 4R2, Canada

**Keywords:** ADHD, psychosocial treatment, psychological interventions, school-based interventions, physical exercise, mind–body intervention, caregiver interventions

## Abstract

Multiple psychosocial interventions to treat ADHD symptoms have been developed and empirically tested. However, no clear recommendations exist regarding the utilization of these interventions for treating core ADHD symptoms across different populations. The objective of this systematic review and meta-analysis by the CADDRA Guidelines work Group was to generate such recommendations, using recent evidence. Randomized controlled trials (RCT) and meta-analyses (MA) from 2010 to 13 February 2020 were searched in PubMed, PsycINFO, EMBASE, EBM Reviews and CINAHL. Studies of populations with significant levels of comorbidities were excluded. Thirty-one studies were included in the qualitative synthesis (22 RCT, 9 MA) and 24 studies (19 RCT, 5 MA) were included in the quantitative synthesis. Using three-level meta-analyses to pool results of multiple observations from each RCT, as well as four-level meta-analyses to pool results from multiples outcomes and multiple studies of each MA, we generated recommendations using the GRADE approach for: Cognitive Behavioral Therapy; Physical Exercise and Mind–Body intervention; Caregiver intervention; School-based and Executive intervention; and other interventions for core ADHD symptoms across Preschooler, Child, Adolescent and Adult populations. The evidence supports a recommendation for Cognitive Behavioral Therapy for adults and Caregiver intervention for Children, but not for preschoolers. There were not enough data to provide recommendations for the other types of psychosocial interventions. Our results are in line with previous meta-analytic assessments; however, they provide a more in-depth assessment of the effect of psychosocial intervention on core ADHD symptoms.

## 1. Introduction

Attention-deficit/hyperactivity Disorder (ADHD) is primarily defined by the presence of developmentally inappropriate levels of inattentive and/or hyperactive-impulsive symptoms, lasting for at least 6 months, occurring in different settings and first manifesting in childhood [1,2]. The presentation of the disorder is characterized by inattention, hyperactivity or both [3].

Despite the clinical validity of the different presentations (or subtypes) of ADHD (i.e., primarily inattentive, primarily hyperactive/impulsive, mixed), these presentations are each associated with heterogeneous trajectories, such that none of them can be specifically associated with different long-term prognoses [3]. Indeed, a wide array of negative outcomes has been associated with ADHD, regardless of the presentation of the symptoms. Children with ADHD present poorer life satisfaction and poorer quality of life than their peers, and this difference persists into adulthood [4]. Individuals with ADHD experience academic underachievement, employment problems, lower income, and poorer health outcomes compared to individuals without ADHD [5]. Not surprisingly, school performance tends to be lower for ADHD children, and they tend to obtain their diploma later and dropout at a higher rate [6,7]. ADHD symptoms negatively affect emotional regulation, which in turn can lead to interpersonal problems [8,9]. Moreover, children with ADHD tend to show deficits in social skills and the processing of social information impairing their ability to integrate into groups of peers [10]. Children and adolescents with ADHD are more at risk of initiating socially deviant behavior such as bullying [11] or, later in life, to commit crimes [12]. Of concern, ADHD is associated with increased all-cause mortality [3]. Adults with ADHD face an increased risk of death by suicide, homicide, and unintentional injuries [13]. People with ADHD are at increased risk of asthma, obesity, diabetes mellitus, allergy, hypertension, sleep problems, psoriasis, sexually transmitted infections, immune disorders, and metabolic disorders [14]. In addition, all-cause mortality is increased in individuals with ADHD [14].

In sum, all presentations of ADHD are associated with impairments that can affect educational, emotional, and interpersonal and health domains, and worsen risks to health outcomes. Given the relatively high incidence of ADHD, this condition is a central concern for health and education authorities. Across different epidemiological cross-cultural studies, it is estimated that the pooled rate is between 4 and 7% of children [3,15] and between 1.4 and 3% of adults [15,16] live with ADHD. This is consistent with the observation that the severity of symptoms tends to diminish with age, even though significant impairment persists for about half of affected individuals. Exact incidence of ADHD is unknown [17]. The meta-analytic studies assessing prevalence reported above were conducted using DSM-IV criteria. The prevalence of ADHD according to DSM-5 criteria is likely slightly higher, both for children [18] and adults [19], since DSM-IV criteria required 6 symptoms for the diagnosis of each subtype while DSM-5 requires only 5 symptoms in adults.

### 1.1. Etiologies of ADHD

A substantial and still growing literature supports the status of ADHD as a neurodevelopmental disorder that can be understood, at least in large part, by biological mechanisms. Numerous studies have shown that ADHD demonstrates high heritability and specific genomes have been linked to the attention or hyperactive symptomatology [14,20,21,22,23,24]. Brain structure, connectivity, activity and neurotransmission patterns of individuals with ADHD tend to demonstrate slight but reliable differences with that of non-ADHD individuals [25,26,27,28,29,30,31,32,33]. ADHD is also reliably associated with decreased performance on certain cognitive tasks (generally involving processing speed, working memory, and attentional tasks) but also with particular types of cognitive errors such as rule violation, omission and commission [34,35,36,37].

In addition to innate neurological development, other factors may contribute to the persistence or even development of ADHD [38,39]. The most comprehensive and current expert consensus [14] is that the development and severity of ADHD symptoms may be influenced by a wide array of psychological, environmental or social factors such as deprivation, stress, family discord, poverty, trauma or exposure to environmental toxins [40]. In addition, it is well accepted that there exist reciprocal, albeit insufficiently understood, interactions between these different factors. For example, innate child impulsivity may generate more hostility from parents, but it also appears that the hostility of parents may later increase the intensity of the child’s symptomatology, in a complex cycle of socio-genetic interaction [41]. As a result of the complex nature of this condition a multipronged and multidisciplinary therapeutic approach should be the mainstay of treatment.

### 1.2. Pharmacological Treatment of ADHD

The use of medication is a cornerstone of the treatment of ADHD. There exists robust evidence supporting the safety and efficacy of these treatment to reduce ADHD symptoms in children, adolescents, and adult populations [42,43,44]. Medication can also reduce related emotional symptoms associated with ADHD [45,46]. Importantly, observational studies with large samples of ADHD participants have also shown that stimulant medication increases school performance [47,48,49] and quality of life (among children and adolescents, but not among adults; [50]). ADHD medication is also associated with reduced risk of depression [51], suicide [13,52], teen pregnancy [53], substance abuse [54], children and adolescent accidental injuries [55,56,57], adult motor accidents [58,59], as well all-cause mortality [60]. Nevertheless, negative attitudes coupled with inexact beliefs are frequent regarding the use of medication for the treatment of ADHD [61]. Such attitudes may be shared by parents [62,63], teachers [64] and even school medical staff [65]. Unfavorable media coverage likely contributes to the hostility against the pharmacological treatment of ADHD [62,66,67].

Furthermore, medication may not be the treatment of choice for all individuals. Not all respond to pharmacological treatment [46] and adverse effects such as reduced appetite, abdominal pain, reduced sleep, hypertension and headaches [68,69] may preclude the use of medication in some individuals. In children, stimulants can delay gains in height and weight [70]. Therefore, despite availability of generally safe and efficacious medications, some individuals with ADHD or their caregivers may prefer non-pharmacological interventions.

### 1.3. Psychosocial Interventions for ADHD

Non-pharmacological interventions or psychosocial treatment should also be considered as a crucial adjunct treatment which may enhance response to pharmacological treatment. The term psychosocial refers to an individual’s psychological development and interaction with the social environment. They involve psychological and social treatment. A variety of psychosocial interventions have been developed or adapted to address ADHD symptomatology in children, adolescents, and adults. They may constitute a primary or adjunctive treatment, and may be based on cognitive-behavioral therapy, mind–body regulation (e.g., physical activity, mindfulness), parent-training, school-based settings or even community settings [71]. The psychosocial interventions for treatment of ADHD are well recognized in the literature. Despite a large body of evidence suggesting that ADHD is primarily caused by biological factors, this does not preclude the contribution of psychological, social and ecological factors on symptoms and their consequences [14]. Indeed, it has been argued that optimal treatment with medication should integrate psychological and social factors for understanding, preventing and, importantly, *treating* illness [72,73,74]. Hence, non-pharmacological aspects of the treatment of ADHD should be thoroughly investigated [75].

Currently, despite the burgeoning literature on psychosocial intervention for ADHD, there is little clear guidance regarding the role of psychosocial approaches in the treatment of ADHD that is based on a systematic review of the literature and uses a well-established method of evidence assessment (e.g., GRADE; [76]). There is a need for a comprehensive and up-to-date summary of psychosocial interventions treatment for ADHD, so that efficacious interventions and population-specific response to each of them may be identified.

Such guidance is necessary for a multiplicity of reasons. As previously mentioned, the choice of psychosocial treatment may be favored because of negatively biased representations of ADHD medication in the media and the general public [67]. Adverse effects may limit the use of medication and may be inefficacious or partially efficacious for some people. It should not be forgotten that psychological and social intervention can also produce iatrogenic effects [77], which may be greater in younger individuals [78]. Furthermore, psychosocial interventions generally require an important investment in resources, both from the treated individuals (and their caregivers) and from the institutions offering the service. Such investment may be more costly for economically disadvantaged populations that, critically, appear to be at greater risk for ADHD [79]. As a whole, it is important for individuals with ADHD, their caregivers and the professionals assisting them to be cognizant of the efficacy of psychosocial interventions for the treatment of core ADHD symptoms in order to make informed therapeutic choices.

The goal of this review and meta-analysis is to explore the existing evidence base regarding psychosocial treatment in order to generate recommendations regarding their use for the treatment of ADHD core symptoms.

ADHD is a highly comorbid condition (with an incidence of 60% to 100%, depending on estimates; [80]) and treatment of ADHD associated with comorbidity will be the object of a subsequent review.

### 1.4. Present Study

The objective of the present study is to systematically review recent (2010 and later) evidence regarding the efficacy of diverse psychosocial interventions on core ADHD symptoms and adverse effects and to compute effect sizes for each type of psychosocial treatment. We divided psychosocial intervention into five categories: Cognitive-behavioral therapy; Caregiver (parent) training; Metacognitive or school-based training; Physical (or mind–body) intervention; and psychosocial intervention. The second objective is to review previous meta-analytic work and to compare the evolution of the recommendations across time and in relation to the quality of evidence. Meta-analytic reviews, despite their systematic approaches, are limited by the criteria and specific approach of the reviewers. By including a comparison with previous work, we hope to facilitate informed and unbiased clinical decisions.

## 2. Methods

### 2.1. Eligibility Criteria

We designed the inclusion and exclusion criteria based on PICO (population, intervention, comparison and outcome) articles published in English from 2010 to 13th February 2020, which investigated populations of individuals of all age groups, both males and females with a clinical diagnosis of ADHD. The intervention is the type of psychosocial treatment used in the studies. The interventions include: Psychoeducation; Therapeutic alliance/Health professional (Therapist) factors; Motivational Interviewing; Cognitive Behavioral Therapy (CBT, Mindfulness-based CBT); Psychotherapy (e.g., Dialectical Behavior Therapy; Trauma-focused; Emotion-focused; Eye movement desensitization and reprocessing (EMDR); Play-based therapy; Social Skills Training; Behavior Therapy (e.g., Behavior Management/Modification, Reinforcement Schedules); Parent/Caregiver training; Family interventions; School-based interventions/accommodations; Workplace interventions/accommodations; Mind–body interventions (e.g., Yoga, Mindfulness-based interventions, Meditation, Relaxation); Healthy lifestyle management (sleep, nutrition, exercise); Coaching (e.g., daily activity scheduling and organization) EF Training/Remediation; and E-therapies

The interventions exclude: Peer-support/Tutoring/Mentoring, Cognitive training; Biofeedback; Nutritional supplements; Diet restrictions; and Deep brain stimulation.

Comparison includes standard care (e.g., CBT), placebo, and no intervention, but excludes Non-standard care

Outcomes to be assessed over the short (<1 year) or long-term (>1 year) and by age (children under 11), adolescents (12–21), adults 21+.

Primary outcomes include core ADHD symptoms and serious adverse events.

They exclude measurements not previously validated. Note that we also included meta-analyses, as one of the aims of the review was to document the evolution of recommendations from older meta-analytic work and to compare the recommendation to that of those we would identify from the present meta-analysis.

We excluded articles not related to ADHD/ADD in humans, that pertained only to pharmacological treatment, that were not a meta-analysis, randomized or otherwise controlled study, those in which ADHD diagnosed was only based on scale scores or if there was a high level of severe comorbidity (more than 15%) of the sample.

For more detailed information about the inclusion and exclusion criteria, please refer to Appendix A.

### 2.2. Information Source

Records were searched on PubMed, PsycINFO, EMBASE, EBM Reviews and CINAHL on 13 February 2020. We were interested in high-quality evidence so that only peer-reviewed articles were considered.

### 2.3. Search Strategy

We combined the filters of three advanced searches. The syntax is reported in Appendix A.

### 2.4. Selection Process

Records were separated in five different intervention categories:Cognitive-behavioral therapy;Caregiver (parent) training;Metacognitive or school based (i.e., teacher training) interventions;Physical exercise or mind–body interventions;Other psychosocial interventions

First, all records were assessed based on their title and abstracts by two independent reviewers (SE and GG). Non-relevant studies were excluded, while potentially relevant studies were grouped into one of the following intervention domains: Cognitive-behavioral, Caregiver (parent) training; Metacognitive or school-based interventions; Physical or mind–body interventions; and social/community-based interventions. Data was extracted from the included records. Eleven reviewers were divided into four dyads and one triad. Each member of the groups independently assessed the inclusion based on full-text screening and validated the extracted data. Disagreement at any point during the selection process was resolved through discussion with all reviewers during periodic meetings.

### 2.5. Data Items

All outcomes related to a validated ADHD symptoms scale, whether it was: (1) self-reported, (2) parents (or caregivers) reported, (3) teachers reported, (4) administered by a healthcare professional or (5) other informant. Furthermore, we extracted demographic data on the participant samples, characteristics of the intervention(s) (type of intervention, duration, means of administration) and of the control intervention, information regarding randomization and blinding procedures, as well as the following methodological data: (1) Was the study preregistered? (2) Was there any evidence of selective reporting? (3) Were the sample sizes reported in analyses consistent with the sample size presented in the methods? (4) If there are exclusions, are they justified?

For meta-analyses, we also extracted data relative to: (a) inclusion and exclusion criteria based on PICO Population, intervention, comparison and outcome; (b) standardized effect sizes from any analysis relating to CORE ADHD outcome; (c) risk of publication bias; and (d) conclusions and recommendations.

### 2.6. Study Risk of Bias

Risk of bias was assessed as part of the GRADE assessment procedure [81]. In accordance with GRADE guidelines, all studies being RCTs, they were attributed a baseline score of 4 (highest possible score). One or two points were subtracted if the study presented limitations regarding: (1) allocation concealment; (2) blinding; (3) data loss at post/follow up coupled with la lack of intent to treat analyses or lack follow up measure; (4) selective reporting; and (5) any other potential source of bias or limitation that could limit the confidence in the results. We did not use the Cochrane tool for assessing the risk of bias since we were able to grade each study.

### 2.7. Effect Measures

For each controlled study, sample size, effect sizes of difference between intervention and control and/or medication groups were extracted either directly or through means and SDs/SEs/confidence intervals, *t-*statistics, *f-*statistics, or *p*-value. For meta-analyses, any summary measures related to comparison between psychosocial intervention and control intervention were extracted. We favored mean gain change when possible.

### 2.8. Synthesis Methods

Effect measures and characteristics of each study (PICOS) and any relevant item regarding the reporting of the results (see Data items) were entered in an Excel file for each review dyad. For each dyad, the records relating to their intervention group were split in half, so that one reviewer would enter the data and the other would verify its validity for one half of the records, and then the role was reversed for the other half. This way, all data were entered and checked by two independent reviewers.

All effects sizes were computed with all statistics and figures computed and created with Rstudio (R version 4.1.3) package “esc” (PBC, Boston, MA, USA) [82]. When data were missing, authors were contacted, or if possible, means and confidence interval were extracted from figures using WebPlotDigitizer (https://automeris.io/WebPlotDigitizer/, accessed on 14 April 2020).

Characteristic of individual studies and meta-analysis were reported in separated tables. Because each study reported several effect sizes (multiples outcomes, multiples assessors, IIT or completer sample, post-test or follow up, effect size of mean change vs. post-intervention mean only), three level meta-analyses were fitted [83]. Typical meta-analyses are based on a two-levels structure (level 1 = global effect size, level 2 = individual studies effects), which cannot adequately reflect multiple dependent effect sizes. When facing multiple effect sizes per study, the researcher must average the effects, or chose only one of the effects and thus critical information may be lost. However, three-level meta-analyses can properly model multiple dependent effect sizes without inflating the effect size, because individual effect sizes are nested in the study level [83]. Traditional multi-level approach or structural equation modeling may be used to create a three-level structure [84].

In the present review, traditional multi-level analyses were conducted, using the “Metafor” [85] package in R (Vancouver, BC, Canada), with a Restricted Maximum Likelihood method. The code used is reported by Assink and Wibbelink in 2016 [86].

This statistical method produced: (1) a pooled effect size with its 95% confidence interval and *p*-value; (2) the *Q* statistic for heterogeneity across the model and its *p*-value [87]; (3) a Likelihood-Ratio-Test of the fit of the three-level model v.s. the two-level model [85]; and (4) the distribution of the *I^2^* heterogeneity statistic across the models’ levels [85].

To explore the possible cause of heterogeneity, a meta-regression was conducted. Sample characteristics (mean age, % male, medication), risk of bias, control comparison, follow up vs. post-measure and mean gain effect size compared post only effect sizes were included as moderators in the three level meta-analysis of the overall results.

We were also interested in summarizing the results of previous and older meta-analytic review so as to present a complete and nuanced view of the literature of psychosocial intervention for ADHD, as well as to facilitate the comparison of our results with that of previous meta-analytic work. We therefore conducted an analysis using effect size reported in the meta-analyses identified with our search strategy. A four levels model was fitted, to account not only for the dependencies at the study level (i.e., one study reported in different comparison in the same MA, or one study being included in different MA), but also for the dependencies at the systematic review level (i.e., all studies included in a single MA share characteristics based on the systematic review methodology). Note that to prevent the redundancy of information that would bias the estimates, studies that were included in multiple already published MA were randomly assigned to only one MA.

### 2.9. Risk of Bias across Studies

To assess publication bias, a meta-regression was conducted, with precision (effect size’s standard error) as a single moderator. In other words, Egger’s test [88] was performed (for another example of a three-level meta-analysis fitting Egger’s test [89]).

### 2.10. Certainty and Recommendation

Certainty of the results was established following GRADE working group guidelines [76], that is, using a combination of (1) overall risk of bias in included studies and for each sub-analyses; (2) inconsistencies (between-study heterogeneity and whether it was explained or not by the different sub-analysis; coherence between present results and past meta-analytic work, coherence between studies included in the meta-analyses and the remaining studies that could not be included for lack of statistic); (3) imprecision of estimates (i.e., large confidence intervals); and (4) risk of publication bias. Recommendations were based on GRADE guidelines [90].

## 3. Results

Thirty-one studies were included in the qualitative synthesis (22 RCT, 9 MA) and 24 studies (19 RCT, 5 MA) were included in the quantitative synthesis. PRISMA Flowchart of the included studies. (Overall effect level, study level, individual effect sizes level, see Figure 1 below).

### 3.1. Study Characteristics and Qualitative Summary

Characteristics of all the RCT included studies are described below in Table 1.

### 3.2. Individual Study Bias

Evaluation of certainty in relation to individual study bias is reported in Table 1. Overall, studies presented severe methodological or reporting limitations. Only eight studies properly conducted and reported the randomization and randomization concealment procedures. Half of the studies used an active control group, only two studies used double blind procedures and 11 studies had no blinding procedure at all or did not report it. Twelve studies did not conduct intent-to-treat analysis, in presence of attrition. However, there was no evidence of selective reporting.

### 3.3. Three-Level Meta-Analyses Results

Forest plots (Figure 2, Figure 3 and Figure 6) summarize the results of different meta-analyses that were conducted. Note that each row represents the results of a three-level meta-analysis, not the results of a single study; that the *n* statistic refers to the total number of effect size included, not the population; and that K refers to the number of independent studies included in each analysis. Standardized effect size *g,* with its 95% CI, are reported with its corresponding *p*-value. The *p*-value for the *Q*-test of heterogeneity is reported (significant value indicates presence of heterogeneity), as well as the I2 statistics for each level (level 1 = sampling error, level 2 = within-study variance, level 3 = between-study variance). Additionally, the three-levels models were compared with classical two-level models. Akaike information criterion and Bayesian information Criterion BIC [123], reported in the Appendix A, almost systematically favored of the three levels model as opposed to the two-level models. Only the *p*-values for the log-likelihood-ratio test comparing the fit of the models are presented in the forest plots. A significant *p*-value indicates a better fit for the three-level model [86].

Note, however, that this log-likelihood-ratio test is helpful in determining the confidence in the result, but that a non-significant result should not be taken as an indication that the two-level models should be preferred. Multiples measurements within a single study must always be considered as nested in the study level [124]. Not doing so would artificially inflate the effect size [84]. Thus, results from the two-level models are never reported, unless the three-level had no convergent solution. Furthermore, note that in absence of overall heterogeneity (i.e., non-significant *Q*-test), it is expected that the model will not provide good fit.

#### 3.3.1. Overall Three-Level Meta-Analyses Results

The first set of analyses (Figure 2) shows that psychosocial interventions have a moderate positive effect on core ADHD symptoms. The pooled effect size was 0.65 (CI = 0.45; 0.85, *p* < 0.001). Log-Likelihood-Ratio test for model fit is generally significant except when there is no evidence for heterogeneity, supporting the relevance of the three-level model. A moderate to large effect is statistically significant for all populations except for preschoolers, for all types of assessors except for teachers’ ratings, and for all outcomes. Between-study heterogeneity (Level 3 *I^2^*; Figure 4) is low to acceptable, except for the preschooler population and for outcomes assessed by teachers, where all the observed variance subsumed by sampling error (i.e., Error *I*^2^).

#### 3.3.2. Three-Level Meta-Analyses Results for Cognitive Behavioral Therapy

For the subgroup analyses of the Behavioral Cognitive Therapy, the three-level structure does not appear to explain results better than a classical two-level model would, as demonstrated by the fact that fit test (log likelihood ratio) is generally non-significant. This is likely attributable to the low level of heterogeneity, as demonstrated by Q-test and the minimal between-study (level 3) heterogeneity (Figure 5).

Thus, the results of all studies converge toward a moderate-high effect on global measures of ADHD symptoms, inattention, as well as hyperactivity/impulsivity symptoms, for self-rating as well as for clinician ratings. Note, however, that the only population studied was adult, as no studies assessed the effect of CBT on children, preschoolers or adolescents.

#### 3.3.3. Three-Level Meta-Analyses Results for Caregiver Intervention

For Caregiver interventions subgroup analyses, the three-level structure generally provides better fit than the two-level structure, as indicated by the “Fit *p*-value” statistics (Figure 6), but it is not the case for the studies with preschoolers, hyperactivity/impulsivity outcomes and observations based on teacher ratings. As can be seen in Figure 7, this could be explained by the very little amount of between-study (level 3) and within-study (level 2) heterogeneity. For these analyses, the three-level model does not offer any advantage over traditional two-level model.

Overall, caregiver interventions demonstrate a moderate effect, The pooled effect size was 0.64 (CI = 0.29; 0.99, *p* < 0.001), although the between-study heterogeneity is substantial. The intervention effect appears homogenous across all ADHD symptoms outcomes, but not across populations. School age children seem to reap great benefits (large effect) from the intervention, whereas the effect for preschoolers is negligible and non-significant. Interestingly, there appear to be no heterogeneity for the preschooler analysis, meaning that all four pooled studies report a similar lack of effect. On the contrary, the children analysis demonstrates significant heterogeneity, which appears mainly explained by within-study variation (level 2 *I*^2^, Figure 7).

The source of this variability is likely the type of assessor: the 37 observations from 10 studies using parent ratings demonstrate a moderate and significant effect, whereas the 19 observations from 3 studies using clinician and the 22 observations from 5 studies using teacher ratings yield a very small and non-significant effect.

#### 3.3.4. Three-Level Meta-Analyses Results for School Based and Executive Intervention

Only two RCTs investigated the effect of School based or executive training intervention. Unsurprisingly, given the very low number of studies, the three-level model did not provide a good fit (*p* = 0.849), and it was not possible to conduct sub-analyses with different populations. Because the outcome studied was different in both studies, it was also not possible to conduct outcome specific analyses. The pooled effect size was 0.53 (CI = 0.32; 0.74, *p* = 0.001), with no evidence of heterogeneity (*p* = 0.51), with 95.54% of the variance explained by sampling error and the rest by within-study variability.

### 3.4. Attrition and Adverse Effects

Very few measures of adherence and acceptance were reported, and no study reported measures of adverse or iatrogenic effects. Thus, we compared the number of participants in baseline vs. post treatment across groups as a proxy for adherence measure. We converted effects from a 2 × 2 contingency table into a single hedge’s *g* for each study. A classical two-level random effect meta-analysis revealed that attrition level was similar in control and intervention groups, yielding a combined non-significant effect size of 0.10 (95% IC: −0.39 to 0.17; *p* = 0.44), with little heterogeneity (*Q* = 0.014), *I*^2^ = 44.80%. Thus, attrition level of intervention groups was similar to that of controls groups across all intervention types.

### 3.5. Meta-Regression

Meta-regressions were conducted with risk of bias (GRADE rating), intervention duration, follow up or immediate post intervention measure, % of medicated sample, % of male and age as moderators. However, the meta-regression model was not significant (*p* = 0.475), with none of predictors reaching significance (*p*-values higher than 0.20).

### 3.6. Publication Bias

There was evidence of publications bias. Meta-regression with precision (standard error) as single predictor reached significance *F*(1170 = 214.03, *p* = 0.012). The coefficient was positive (*B* = 1.18, *p* = 0.01), indicating that the larger the standard error was, the larger the effect. Likewise, Log Rank test showed a moderate association between error and effect sizes (Kendall’s tau = 0.32, *p* < 0.0001).

Two additional meta-regression with precision as a single predictor were also conducted on CBT studies and Caregiver intervention studies, to separately assess the publication bias (other intervention domains were not powered enough to enable the analysis). There was no evidence of publication bias in CBT studies *F*(1,52) = 0.07 (*p* = 0.786), but there was for caregiver, as precision was a positive predictor of the effect size *F*(1,70) = 15.52, *p* = 0.001.

### 3.7. Four-Levels Meta-Analysis

The results of the four-level meta-analysis, which include data from previous meta-analytic work, are presented in Table 2. Only three separate analyses could be conducted: an overall analysis, parent rating and teacher rating. Effect sizes are similar to those obtained in the present review and teacher/educator’s ratings also are not significant. Note that the overlap between studies included in all the meta-analyses and the present meta-analytic work were very low (14%). In Table 3, Error I2 refers to the amount of total variation explained sampling error, whereas level 2 I2 and level 3 I2 refers to the amount of total variation explained by within-study and between-study variance, while level 4 I2 refers to the variance between the included meta-analyses.

### 3.8. Certainty of Assessment and Recommendation

GRADE-style Tables, with recommendations, are presented for all results (Table 4) as well as for each intervention (Table 5, Table 6, Table 7 and Table 8). The recommendations follow GRADE suggestion:

“Do it” or “don’t do it”—indicating a judgment that most well-informed people would make;

“Probably do it” or “probably don’t do it”—indicating a judgment that a majority of well-informed people would make but a substantial minority would not ([90] p. 1493).

Note that in our evaluation of the trade-off of the intervention, only the time required was considered, because no adverse effects were reported and analyses showed similar attribution for both experimental and control group (see Section 3.4), suggesting little harm associated with psychosocial intervention. Nevertheless, the potential delay in accessing pharmacological treatment is an often-neglected potential harmful effect.

## 4. Discussion

This study took advantage of recent advances in meta-analytic statistics to summarize and quantify results from 147 effect sizes, pooled from 19 RCTs. The three-level meta-analyses showed that psychosocial interventions had a moderate effect on core ADHD symptoms. Importantly, the three level models generally show better fit than the two-level models, except when overall heterogeneity was very low. This is to be expected, because with low heterogeneity there is no additional variance that a three-level model can explain [86].

Furthermore, our results were consistent with past meta-analytic work. However, in comparison of previous meta-analyses and systematic reviews, the present results allowed for a more granular comprehension of the effect of psychosocial intervention, as well as in-depth assessment of the quality of evidence with a qualitative analysis, where three additional RCTs and four additional meta-analyses complemented the quantitative analyses. Thus, several recommendations emerge from the present review.

### 4.1. Psychosocial Intervention for ADHD: Which One and for Whom?

Overall, it appears that psychosocial interventions are promising and efficacious treatments for improving core symptoms of ADHD. However, existing interventions may not benefit all populations, or existing data may be insufficient to confidently support the benefits of a given intervention for a specific group or a given outcome. Therefore, our recommendations are intervention and population specific, as recommended by GRADE [76,90].

We recommend Cognitive Behavioral Therapy for reducing global, inattention and hyperactivity/impulsivity symptoms of ADHD for adults. Our confidence in the efficacity of this intervention is moderate;We recommend caregiver interventions for reducing inattention and hyperactivity/impulsivity symptoms of ADHD of school age children. Our confidence in the efficacity of this intervention is low;We recommend caregiver interventions for reducing inattention and hyperactivity/impulsivity symptoms of ADHD of school age children. Our confidence in the efficacity of this intervention is very low;We recommend School based or Executive interventions for reducing inattention and hyperactivity/impulsivity symptoms of ADHD of school age children. Our confidence in the efficacity of this intervention is very low;We *do not* recommend caregiver interventions for reducing inattention and hyperactivity/impulsivity symptoms of ADHD of preschoolers. Our confidence in the *lack of efficacity* of this intervention is low;No recommendation can be formulated for other interventions, populations, and outcomes (e.g., disorganization symptoms)

We did not provide individualized recommendations for separate outcomes, because the effect of all interventions appeared homogenous across different core ADHD symptoms (note, however, that few studies reported outcomes related to deficits in organization). Inversely, the type of assessor completing the ADHD ratings was an important source of variability. For non-adult populations, multiple analyses confirmed that teacher and educator ratings were much lower than that of parents, clinicians or self-rated assessment. This result was coherent with previous meta-analytic work (Table 4). Therefore, there can be high confidence that the assessments of teachers are generally lower than other assessors, especially parents, following a psychosocial intervention for ADHD.

It is well-established that parents and teacher reports of ADHD are poorly correlated [128,129]. However, parents’ ratings are generally more severe than that of teachers [130,131]. Furthermore, the present meta-analysis only included controlled clinical trials, such that any systematic assessment biases (whether positive or negative) should be accounted for by the control group. Yet, the present results demonstrate that parents report more symptoms improvement than the teachers.

While surprising, such results do reproduce the same measurement disparity that is observed in pharmacological trials of ADHD. Clinical studies assessing the effect of stimulants have also shown that the improvement ratings of teachers were lower than those of parents, clinicians or children’s self-assessment [43]. The rating instability could be influenced by baseline difference in ADHD assessment (i.e., if teachers are more conservative in their initial assessment, there is less room for improvement), but it appears that there is a true discrepancy in ratings the evolution across time of ADHD symptoms following a pharmacological treatment [132].

Several hypotheses have been proposed to explain this difference in ratings, such as measurement error in the scales, context-dependent rating of the behaviour, assessor bias, influence of demographic variables on the assessor’s perception of the behaviour [133,134,135,136]. While the present meta-analytic work was not designed to address the causes of this inconsistency in measurement between parent and teacher, it is worth nothing that clinician ratings (for Caregiver interventions, an intervention that mainly targets school age children) are also low and not statistically significant. This finding supports the hypothesis that teachers may be less biased than parents [134] and is one of the main reasons why we downgraded the confidence in the strength of evidence for Caregiver intervention.

No other demographic variable was considered in our recommendation, as the meta-regression revealed that none of or hypothesized predictors of the effect size were significant. Importantly, GRADE assessment [81] of individual study quality (risk of bias) was not a significant predictor of the effect size. This means that the numerous biases identified in the included studies might not have that much of an impact on the results [85]. However, such biases were so frequent that an alternative explanation may be that high-quality studies were too few to influence the moderation.

### 4.2. Limitations of the Present Review

While this systematic review had simultaneously a large scope and a granular approach that provided multiples detailed summaries combined with in-depth strength of evidence assessment, it remains an incomplete picture of the effects of psychosocial intervention on ADHD. A first limitation to consider is that no secondary outcomes were included in the analyses. While core ADHD symptoms are an important source of impairment for individuals with ADHD, many other factors can influence treatment outcomes. Ultimately, clinicians, parents, educators and individuals with ADHD might be as interested in improving quality of life, quality of relationships, academic performance and emotional regulation as they are in reducing core symptoms. In addition, this population was limited to individuals with low levels of comorbidities in an attempt to delineate the effect of interventions on core symptoms. However, most individuals with ADHD do present comorbidities [80] and the effect of such interventions on comorbidities will need to be the focus of future reviews.

### 4.3. Future Research

While the present systematic review and meta-analysis constitute an important step in comprehension of the non-pharmacological treatments of ADHD, one of the main findings is that the methodological quality of RCTs studying psychosocial interventions for ADHD is generally poor. We urge researchers to:Implement robust randomization procedures, including proper concealment of the allocation sequence [137];Assure that all assessment personnel and participants are blinded. While a double blind paradigm may be difficult to implement in the context of psychosocial intervention, it is possible to design active control group that mimics some aspect of the intervention group, thus creating a credible alternative to the active intervention. This design is referred to as “dual blind” and has been used in other psychosocial research, such as mindfulness-based intervention [138];Preregister the study protocol, so that reporting bias may be correctly assessed [139].

While time and resource consuming, applying these recommendations will ensure that more meaningful conclusions can be gleaned from the available data, ultimately benefitting researchers, clinicians, caregivers and ADHD individuals themselves.

## 5. Conclusions

This systematic review and meta-analysis demonstrated that recent (>2010) psychosocial interventions had a moderate effect on core ADHD symptoms, although this effect was not homogenous across all types of intervention nor across all populations. Making use of advances in meta-analytic statistics, three-level models were fitted to the data, enabling the combination of 147 dependent effect sizes across 19 studies. Additionally, four-level models were also fitted on five previous meta-analyses and showed that present results were consistent with older data using less stringent inclusion criteria. However, there was a lack of studies for some interventions (school based and executive interventions, mind–body and physical interventions) and some population (preschoolers and adolescents), precluding formation of recommendations. Furthermore, because of the high risk of bias in included studies and evidence of publication bias, only a weak recommendation can be made for most of the interventions. Clearly, rigorous, well-designed and controlled studies are needed.

## Figures and Tables

**Figure 1 brainsci-12-01023-f001:**
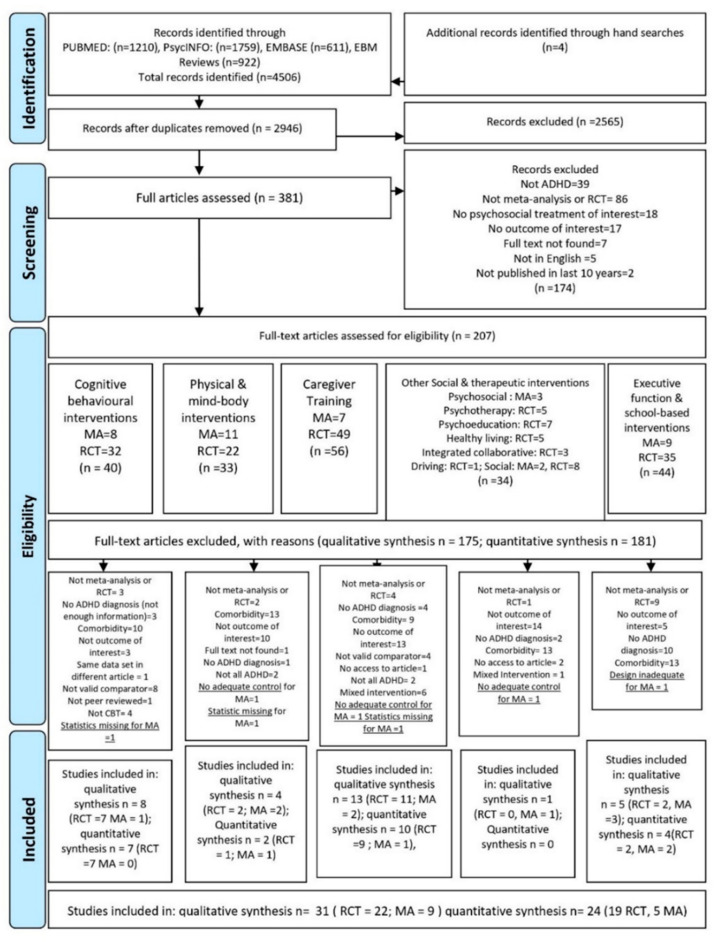
PRISMA Flowchart of the included studies [91].

**Figure 2 brainsci-12-01023-f002:**
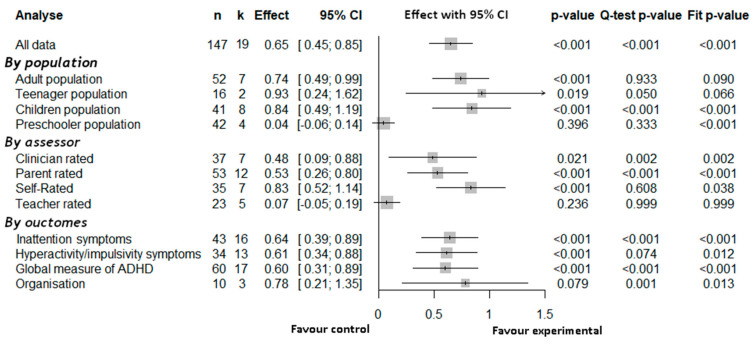
Results of three-level meta-analyses, for all interventions combined. *n* = number of total observations; k = number of studies; Effect = standardized effect size g; 95% CI = 95 confidence interval; *p*-value = traditional *p*-value of the effect; Q-test *p* value = *p*-value of the Q test for heterogeneity (generally, *p* < 0.1 indicates the presence of heterogeneity); Fit *p*-value = results of the likelihood the Restricted Maximum Likelihood-Ratio test comparing the three-level model to a two-level model (*p* < 0.05 indicates a better fit for the three-level model). The models were estimated using REML method.

**Figure 3 brainsci-12-01023-f003:**
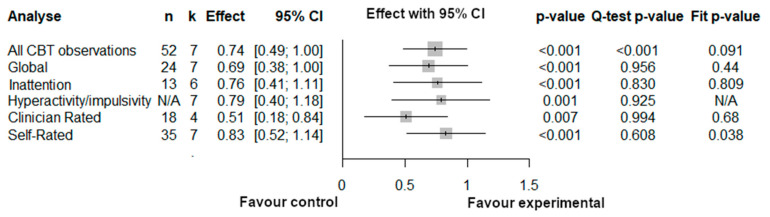
Three level-analyses for Cognitive behavioral therapy. Note that the three-level model for the Hyperactivity/impulsivity did not converge, so a traditional two-level model (with one average of multiple observation per study) was used instead.

**Figure 4 brainsci-12-01023-f004:**
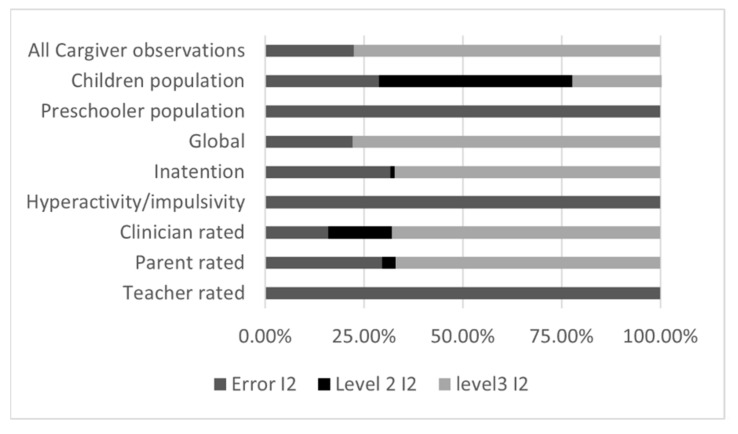
Heterogeneity distribution for three Level meta-analyses, all interventions combined. Error I2 refers to the amount of total variation explained sampling error, whereas level 2 I2 and level 3 I2 refers to the amount of total variation explained by within-study and between-study variance. Note that Cargiver equals Caregiver and inatention equals inattention.

**Figure 5 brainsci-12-01023-f005:**
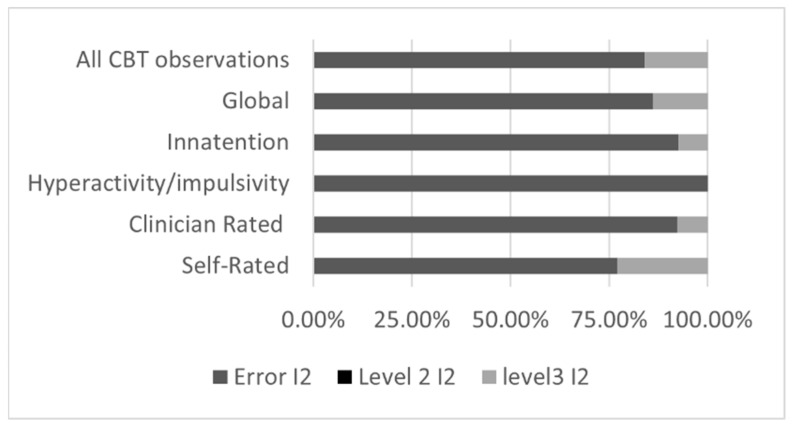
Heterogeneity distribution for three-level meta-analyses, Cognitive Behavioral Therapy.

**Figure 6 brainsci-12-01023-f006:**
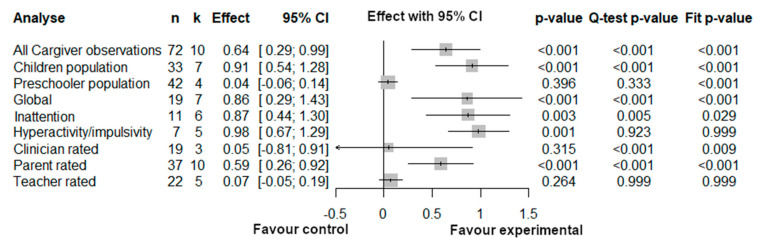
Three level-analyses for Caregiver Interventions.

**Figure 7 brainsci-12-01023-f007:**
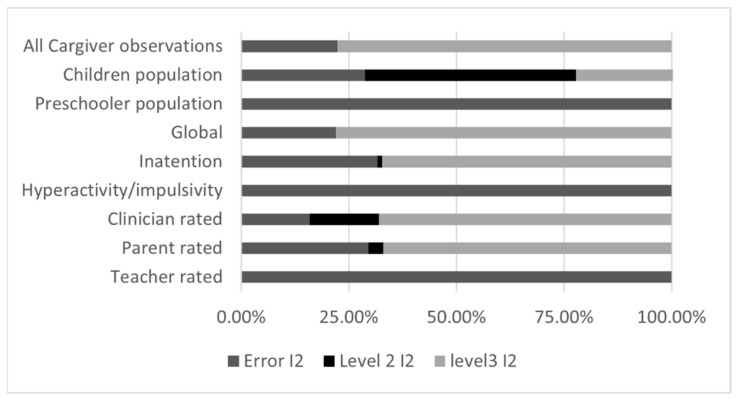
Heterogeneity distribution for three-level meta-analyses, Caregiver Interventions. Cargiver = Caregiver, inatention = inattention.

**Table 1 brainsci-12-01023-t001:** Characteristics of all the RCT included studies ^1^.

First Author	Year	*n* ^2^	Intervention	Population (Age Range If Provided)	Mean Age (% Male)	Comparator	% (Exp, Ctrl) on RX ^3^	Follow Up (Weeks)	ADHD Scales	GRADE Rating (Limitation Domain or Other Reason If Downgraded)
CBT										
Corbisiero [92]	2018	35–41	10–12 weeks individual CBT	Adults (18–49)	32.05 (60%)	TAU	100%, 100%	39	WRAADDS (C), CAARS (S:S; O:L) ADHD-SR (S)	3 (Blinding)
	2018	39–46	16 weeks individual CBT	Adults (18–65)	35.9 (69%)	TAU	63%, 87%	42	CSS (S;O), CGI-I (S,C ^4^)	3 (Blinding)
Emilsson [93]	2011	21–35	15 session group/individual CBT (R&R2ADHD)	Adults	33.9 (37%)	TAU	100%, 100%	12	K-SADS-PL (C), CGI-S (C), BCS (S), RATE-S(S)	3 (Blinding)
Gu [94]	2018	54	6 weeks Individual MCBT	Adults (19–24)	20.3 (55%)	Waitlist	72%, 77%	12	CAARS (S:S)	2 (Allocation concealment, control group, blinding)
Pettersson [95]	2017	28–32 ^5^	10 weeks internet individual and group therapy (In focus)	Adults	34.7 (33%)	Waitlist	43%, 50%	None	CSS (S)	3 (control group, blinding)
Safren [96]	2010	67–84	12 sessions (15 weeks) of individual CBT	Adults (18–65)	43.2 (56%)	Relaxation training and emotional support	100%, 100%	26 and 52	CCS (S), ADHD-RS-IV (S), CGI-S (C)	4
Schonberg [97]	2013	44	12 weeks Group MCBT	Adults (19–53)	37.0 (48%)	Waitlist	48%, 62%	None	CAARS(S:S)	2 (Allocation Concealment, blinding, inactive control group, no ITT)
Solanto [98]	2010	81	Meta-cognitive therapy	Adults (18–65)	41.7 (34%)	Group support	Not reported	None	AISRS-IN (C), CAARS (O:L); BAS (S) ON-TOP (S)	2 (Allocation concealment, blinding, no ITT)
**Physical and Mind body intervention**										
Kang [99]	2011	28	6 weeks (12 90 sessions) of sport therapy	Children	8.5 (100%)	Education on behavior control	100%, 100%	None	K-ARS (P,T)	2 (allocation concealment, blinding)
Meßler [100] *	2018	28	3 weeks HIIT training (3 sessions/w of 4 × 4 min intervals)	Children (8 to 13)	11 (100%)	weeks of low intensity physical activity (1 90 min session/week	36%, 29%	None	FBB-HKS (P) SBB-HKS (S) DISYPS-II (?)	2 (allocation concealment, blinding)
**Caregiver**										
Abikoff [101]	2015	101	New Forest Parenting Program (8 × 60–90 min individual sessions)	Preschooler (3–4)	73%	Waitlist	0%, 0%	None	CPRS-R (P), CTRS-R (T) DuPaul ADHD-RS-IV (C)	4 * Example of dual blind
Bai [102]	2017	89	3 months of medication adherence program, individual and group, online and in person	Children/adolescent (6–16)	9.5 (80%)	General clinical counselling	1%, 1%	None	ADHD-RS-IV (P)	4
Behbahani [103]	2018	48–52	Mindful Parenting programme (8 weeks)	Children (7–12	? (66%)	Waitlist	100%, 100%	8	SNAP- IV (P)	2 (allocation concealment, blinding, passive control group, no ITT)
Haack [104]	2017	128	10–13 weeks Parent Focused Training	Children (7–11)	8.6 (58%)	TAU	9%, 2%	21–30	CSI (P;T) COSS (T;P)	3 (Allocation concealment, blinding)
Herbert [105]	2013	31	The Parenting Your Hyperactive Preschooler program (14 × 90 min session)	Preschooler (34–76 mo)	4.6 (74%)	Waitlist	18%, 7%	None	DBRS (P)	3(blinding, passive control group)
Lange [106]	2018	129–148	New Forest Parenting Program (8 × 60–90 min individual sessions)	Preschooler/children (3–7)	? (73%)	TAU	0%, 0%	36	ADHD-RS-IV (P)	4
Maleki * [107]	2014	36	Barkley Parent Training program	Children (6–12)	? * (?)	Working memory training	100%, 100%^6^		SNAP-IV (P) CLCL (P)	3 (allocation concealment, blinding)
Pfiffner [108]	2014	90–122	12 weeks Parent focused training	Children (7–11) inattentive type only	8.6 (58%)	TAU	15%, 14%	12 weeks	CSI (T;P), COSS (T;P), CGI-I (T;P), CGI-S (T)	2 (allocation concealment, blinding)
Shafiee-Kandjani [109]	2017	25–32	New Forest Parenting Program 8 × 60–90 min individual sessions)	Children (6–12 years)	7.1 (100%)	TAU	??	4	CPRS (P)	4
Sonuga-Barke [110]	2018	173–175	New Forest Parenting Program (12 individual sessions) and Incredible years (12 group sessions)	Preschooler (2 y 9 mo–4 years 6 mo)	3.55 (68%)	TAU	0%,0%	26	SNAP-IV (P,T) DOA (C)	4
Yusuf [111]	2019	48	Triple P program (8 weeks, 5 × 120 m session + 3 × 15–30 m phone session)	Children (7–12)	10.25 (79%)	Waitlist	100%, 100%	None	DuPaul ADHD-RS-IV (C), GCI-S (C)	2 (allocation concealment, blinding, passive control group)
**School Based and Executive**										
Corkum [112]	2019	58	6 weeks Online teacher training	Children (grade 1 to 6)	8.2 (88%)	Waitlist	85%, 76%	6	CPRS-3 (P) CPRS-3 (T) IRS (P,T)	3 (Blinding, passive control group)
Schultz [113]	2017	88–216	Challenge Horizon Program (1 year after school training, 2 × 75 min/week)	Adolescents (Grade 6 to 8)	12.15 (72%)	Community Care	49%, 47%	28	COSS (P) DBRS (P)	3 (blinding)

^1^ Studies excluded from quantitative review are marked with an asterisk (*). Scales were used only for diagnostic purposes, as well as secondary outcomes are their related instruments, are not presented here, but are reported in the Appendix A. ^2^ Ranges of valid observation which meta-analyses were based. May differ from total sample reported in appendix tables because of: (1) participants were excluded after randomization and there were no ITT analyses; (2) observations were missing; or (3) the study had multiple groups which were not all included in the meta-analyses. ^3^ Percentage of the sample (in experimental group, control group) that was on ADHD medication at baseline. ^4^ The CGI scores were not reported, only the % of sample that improved, so GCI scores are not included in meta-analyses. ^5^ Note that only the iCBT with group session was included in the analysis, and not the “self-help version”. ^6^ More specifically, Ritalin (no participants on other medication were included). Detailed Data of Cognitive Behavioral Therapy Studies are available on Appendix A [92,93,94,96,97,98,114,115], Data of Physical and Mind Body Studies on Appendix A [99,100,116,117]. Caregiver Interventions Studies are available on Appendix A [101,103,105,106,107,109,110,111,118,119,120] and Detailed Information of School Based and Executive Studies are available on Appendix A [108,112,121,122].

**Table 2 brainsci-12-01023-t002:** Meta-analyses of previous data synthesized in published meta-analysis, showing a similar overall effect size and the same parent/teacher.

First Author	Year	Number of Studies	Total *n*	Design of Included Studies	Type of Intervention	Population	Average Study Quality ^1^	Evidence of Publication Bias
Bikic [121]	2017	12	1054	RCT	Executive training	Children and adolescents	Low (high risk of bias)	Not reported
Cerrillo-Urbina [116]	2015	8	249	RCT	Physical exercise	Children and adolescents	Low	No
Gaastra [125]	2016	24	Not reported	Whitin-study design	School based intervention	Children	Moderate to high	Yes
Hodgson [126] *	2014	4	206	Unclear	Behavioral training, school based, executive training, parent training	Preschoolers to adolescents	Not reported	Not reported
Mulqueen [118]	2015	8	399	RCT	Parent training	Preschoolers	Not reported	No
Rimestad [127]	2011	16	1003	RCT	Parent training	Preschoolers	Moderate	No
Zang * [117]	2019	14	574	RCT, case-control	Physical exercise	Children and adolescents	Moderate	Yes

* The study was only considered in the qualitative analysis. ^1^ According the authors of the included meta-analysis.

**Table 3 brainsci-12-01023-t003:** Meta-analyses of previous data synthesized in published meta-analysis, showing a similar overall effect size and the same parent/teacher rating disparities.

	All	Parent	Teacher
Observations	69	42	27
Studies	49	35	24
Meta-analyses	5	5	3
Effect Size	0.604	0.524	0.610
Standard Error	0.148	0.066	0.38
*p*-value	0.001	0.001	0.12
Q*p*-value	0	0.001	0.00
Fitvslvl3	0.002	0.0327	0.01
Fitlvl4	0.0001	0.0001	0.00
Error *I^2^*	10.03%	39.90%	3.57%
Level 2 *I^2^*	12.71%	0.00%	0.00%
Level 3 *I^2^*	63.51%	61.10%	55.56%
Level 4 *I^2^*	13.75%	0.00%	40.86%

**Table 4 brainsci-12-01023-t004:** Psychosocial intervention compared to control (active and wait-list) for core ADHD symptoms and for all population.

	Number of Studies	Total *n*	Quality of Evidence (GRADE)	Pooled Effect Size (95% CI)	Recommendation
Overall	20 ^1^	1673	⊕⊕⊝⊝ WEAK (unreported or inadequate allocation concealment, inadequate blinding, lack of ITT analyses evidence of publication bias)	0.66 (0.50; 0.82)	Probably do it
Cognitive behavioral therapy	8	417	⊕⊕⊕⊝ MODERATE (unreported or inadequate allocation concealment, inadequate blinding)	0.74 (0.50; 0.98)	Do it
Mind–body intervention and physical exercise	1 ^2^	56	⊕⊝⊝⊝ Very Weak (Insufficient data)	N/A	No recommendation
Caregiver intervention	10 ^3^	962	⊕⊝⊝⊝ Very Weak (unreported or inadequate allocation concealment, inadequate blinding, lack of ITT analyses in presence of attrition, variability in effect by population and by outcome assessor, publication bias)	0.64 (0.37–0.91)	Probably do it
School-based intervention	2 [118] ^4^	274	⊕⊝⊝⊝ Very Weak (Insufficient data, allocation concealment, inadequate blinding, lack of ITT analyses)	0.52 (0.30; 0.74)	Probably do it
Adults (*note: only based on CBT)*	8 [121]	56	⊕⊕⊕⊝ MODERATE (unreported or inadequate allocation concealment, inadequate blinding)	0.74 (0.50; 0.98)	Do it
Adolescents	2	305	⊕⊝⊝⊝ Very Weak (Insufficient data)	Not enough study for analyses	No recommendation
Children	15 [125]	998	⊕⊕⊝⊝ WEAK (unreported or inadequate allocation concealment, inadequate blinding, lack of ITT analyses in presence of attrition, high between-study heterogeneity)	0.73 (0.49; 0.97)	Probably do it
Preschoolers	5 [119]	455	⊕⊕⊝⊝ WEAK (unreported or inadequate allocation concealment, inadequate blinding, lack of ITT analyses in presence of attrition)	0.32 (−0.01; 0.63)	Probably don’t do it

⊕: positive; ⊝: negative. ^1^ Additionally, three RCTs and four meta-analyses were included and the qualitative assessment. ^2^ Additionally, one RCTs and two meta-analyses were included and the qualitative assessment. ^3^ Additionally, two RCTs and two meta-analyses were included and the qualitative assessment. ^4^ Additionally, three meta-analyses were included and the qualitative assessment.

**Table 5 brainsci-12-01023-t005:** Cognitive Behavioral Therapy compared to control (active and wait-list) for ADHD individual for each population. ? means unable to assess quality of evidence.

	Studies	Total *n*	Quality of Evidence (GRADE)	Pooled Effect Size (95% CI)	Recommendation
Adults	8	417	⊕⊕⊕⊝ MODERATE (unreported or inadequate allocation concealment, inadequate blinding)	0.74 (0.50; 0.98)	Do It
Adolescents	0		? No data	N/A	No recommendation
Children	0		? No data	N/A	No recommendation
Preschoolers	0		? No data	N/A	No recommendation

⊕: positive; ⊝: negative.

**Table 6 brainsci-12-01023-t006:** Physical and mind–body intervention compared to control (active and wait-list) for ADHD individual for each population. ? means unable to assess quality of evidence.

	Studies	Total *n*	Quality of Evidence (GRADE)	Pooled Effect Size (95% CI)	Recommendation
Adults	0	0	? No data	N/A	No recommendation
Adolescents	0	0	? No data		No recommendation
Children	1 (+1 quali)	56	⊕⊝⊝⊝ Very Weak (Insufficient data, allocation concealment, blinding)		No recommendation
Preschoolers	0	0	? No data		No recommendation

⊕: positive; ⊝: negative.

**Table 7 brainsci-12-01023-t007:** Caregiver intervention compared to control (active and wait-list) for ADHD individual for each population. ? means unable to assess quality of evidence.

	Nb of Studies	Total *n*	Quality of Evidence (GRADE)	Pooled Effect Size (95% CI)	Recommendation
Adolescents	1	89	⊕⊝⊝⊝ Very Weak (Insufficient data)	N/A	No recommendation
Children	7	907	⊕⊝⊝⊝ Very Weak (unreported or inadequate allocation concealment, inadequate blinding, lack of ITT analyses in presence of attrition, variability in effect by outcome assessor, publication bias)	0.91 (0.54–1.28)	Probably do it
Preschoolers	4	455	⊕⊕⊝⊝ WEAK (unreported or inadequate allocation concealment, inadequate blinding, lack of ITT analyses in presence of attrition)	0.04 (0.06; 0.14)	Probably don’t do it

⊕: positive; ⊝: negative.

**Table 8 brainsci-12-01023-t008:** School based and executive intervention compared to control (active and wait-list) for ADHD individual for each population. ? means unable to assess quality of evidence.

	*k (j)*	Total *n*	Quality of Evidence (GRADE)	Pooled Effect Size (95% CI)	Recommendation
Adults	0	0	? No data		No recommendation
Adolescents	1	216	⊕⊝⊝⊝ Very Weak (Insufficient data)	N/A	No recommendation
Children	1	58	⊕⊝⊝⊝ Very Weak (Insufficient data)	N/A	No recommendation
Preschoolers	0	0	? No data		No recommendation

⊕: positive; ⊝: negative.

## Data Availability

Data can be supplied through contacting corresponding author.

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
