# Peer review of "Psychosocial Interventions for Attention Deficit/Hyperactivity Disorder: A Systematic Review and Meta-Analysis by the CADDRA Guidelines Work GROUP"

_brainsci, 2022, doi:10.3390/brainsci12081023_

Round 1

Reviewer 1 Report

Minor points

1. The format of the manuscript is not as provided in the ‘‘Instruction for authors’’

2. In the reviewer’s opinion, the first two paragraphs of the introduction should be more appropriately connected with the background to avoid misunderstandings.

3. This phrase can be removed ‘‘We decided not to search nor include gray literature or pre-prints, as we were interested in high quality evidence’’

4. It is advised to remove the word obvious from ‘’Obvious non-relevant studies were excluded’’

Author Response

Dear Reviewer 

Please see the attached letter.

Many thanks

Authors. 

Reviewer 2 Report

Thank you for the opportunity to review this meta-analysis.

At first, I think that this meta-analysis present some flags, such as:

(1) Filters related with publication data (since 2010)

(2) What is the reason for not to search in other sources, like gray literature?

(3)  I consider that the search must be updated, becuase search present in this study was until February 13, 2020.

(4) References do not appear in Brain Sciences style. I consider that authors must be more careful with this aspect.

(5) GRADE assess quality of evidence. Cochrane risk of bias tool is the most indicated to assess the risk of bias, if articles are RCT.

(6) Inclusion criteria are not correct. These must be PICOS more others.

(7) Statistical analysis is desorganized and poor of information.

(8) Figure 1 is a mistake if appears in methods. This is an usual mistake that can be solucionated searching other meta-analyses.

(9) In results, statistical findings must be written. 

(10) This meta-analysis has not been registered in PROSPERO.

I consider that this manuscript must be corrected.

Author Response

Dear Reviewer

Thanks so much for your expert review. Please see the attached letter.

Thanks again.

The authors

Reviewer 3 Report

The authors processed the topic "Psychosocial interventions for Attention Deficit/Hyperactivity disorder:". The issue itself is very important, but I see the main problem in the fact that the contribution does not bring any original knowledge, which the authors themselves state in the conclusion.

“There were not enough data to provide recommendations for the other types of psychosocial interventions. Our results are in line with previous meta-analytic assessments, yet provide a more in-depth assessment of the effect of psychosocial intervention on core ADHD symptoms. ‘

1.     What is it ‘My comprehension is what gets me. … I’ll read an entire page and then finally realize I was thinking about what I had for lunch yesterday, and then I’ll have to go back and start all over again:“ and It was a relief to find out what was wrong, why I got into fights, why I always quit my job, why I always get tired of people, and why I always get so pissed off at everyone (Hansson? Why citation has several authors

2.     A better definition is needed: “This now well-known condition is labeled Attention Deficit / Hyperactivity Disorder (ADHD).“

3.     Add information on quality of life „Children with ADHD have a poorer life satisfaction than their peers, and this difference persists into adulthood (Lee et al., 2016).“

4.     Native-speaker correction is needed

5.     Add information about prevalence and incidence of ADHD

6.  Add and delete information only from the RCT study ‘Pharmacological treatment of ADHD”

7.     More comprehensive information is needed on ‘psychosocial interventions for ADHD’. 

Author Response

Dear reviewer 

Thanks so much for your expert review. Please see the attached letter.

Thanks again

The authors

Reviewer 4 Report

Introduction

1.     Page 2. Authors wrote “My comprehension is what gets me. … I’ll read an entire page and then finally realize I was thinking about what I had for lunch yesterday, and then I’ll have to go back and start all over again (Lefler, Sacchetti, & Del Carlo, 2016). It was a relief to find out what was wrong, why I got into fights, why I always quit my job, why I always get tired of people, and why I always get so pissed off at everyone (Hansson Halleröd, Anckarsäter, Råstam, & Hansson Scherman, 2015)”. I was not able to catch the meaning of these sentences. If these are some thoughts of ADHD patients, this should be clarified.

2.     Page 2. Authors wrote “The prevalence of ADHD according to DSM-5 criteria is likely slightly higher, both for children (Voort, He, Jameson, & Merikangas, 2014) and adults (Matte et al., 2015).”. I suggest that Authors quantify such prevalence.

3.     Since the focus of the study is represented by the psychosocial interventions for ADHD, I suggest that Authors shorten the Introduction section just focusing on this kind of treatment.

4.     Page 4. Authors wrote “there is little clear guidance regarding the role of psychosocial approaches in the treatment of ADHD”. I believe that Authors should expand this sentence highlighting the current state of knowledge about the effectiveness of psychosocial approaches for ADHD.

5.     Page 4. I suggest that Authors describe in detail the psychosocial interventions.

6.     Page 4 and 5. I am not sure that cognitive-behavioral therapy can be considered a psychosocial intervention. On the contrary, I think that it qualifies as a psychotherapy.

Discussion

1.     Page 18. In line with my comment number 6 on the Introduction section, I believe that cognitive-behavioral therapy does not qualify as a psychosocial treatment but as a psychotherapeutic treatment.

Author Response

Dear Reviewer

Thanks so much for your expert review. Please see the attached letter.

Thanks again

The authors

Round 2

Reviewer 2 Report

I consider that this review present bias related with publication date and language filters. These filters can produce risk of publication bias, under or overestimating findings.

On the other hand, search in gray literature can be unified by independent reviewers.

If the study was performed in 2020 and it was published in 2022, it would be interesting to include studies published between 2020 and 2022.

However, other concerns have been correctly resolved.

Author Response

Dear Reviewer

       Thanks so much for you excellent and helpful comments. I appreciate your expertise and input in this important area. Below is the response to your concerns.

       Thanks again

Reviewer 3 Report

1. Please do not choose two different citation style. „Numerous studies have shown that ADHD demonstrates high heritability and specific

genomes have been linked to the attention or hyperactive symptomatology (14, 20,

21, 22, 22, 23, 24,). Brain structure, connectivity, activity and neurotransmission patterns

of individuals with ADHD tend to demonstrate slight but reliable differences with that of

non-ADHD individuals (Boedhoe et al., 2020; L. Chen et al., 2016; Cortese et al., 2012; Curatolo,

D'Agati, & Moavero, 2010; Hoogman et al., 2017; Hoogman et al., 2019; Lukito et

al., 2020; Norman et al., 2016; Rubia, 2018).“

2. Use the information only from the RCT study in this paragraph „Pharmacological treatment of ADHD“

3. Use passive voice no active voice „We designed the inclusion and exclusion criteria based“ in all manuscript 

Author Response

Dear Reviewer

       Thanks so much for you excellent and helpful comments. I appreciate your expertise and input in this important area. Attached are the responses to your concerns.

       Thanks again

         The Authors

Reviewer 4 Report

Authors’ reply to my previous comment number 3 on the Introduction section

1.     Since the pharmacological treatment is outside the aim of the study, I would suggest that Authors delete the paragraph 1.2.

Authors’ reply to my previous comment number 4 on the Introduction section

1.     Despite the different guidelines between countries, I am wondering whether there is a strongest evidence-based psychosocial treatment of ADHD.

Introduction

1.     Authors should revise this sentence in the paragraph 1.4 because psychosocial interventions are listed as a separate category from the CBT and other treatments: “We divided psychosocial intervention in five categories: Cognitive-behavioral therapy, Caregiver (parent) training, Metacognitive or school-based training, Physical (or mind-body) intervention and psychosocial intervention”.

Figures

1.     Figure 1 is not readable.

Author Response

Dear Reviewer

       Thanks so much for you excellent and helpful comments. We appreciate your expertise and input in this important area. Attached is the response to your concerns.

       Thanks again
